# Neuropsychological Characterization of Aggressive Behavior in Children and Adolescents with CD/ODD and Effects of Single Doses of Medications: The Protocol of the Matrics_WP6-1 Study

**DOI:** 10.3390/brainsci11121639

**Published:** 2021-12-11

**Authors:** Carla Balia, Sara Carucci, Annarita Milone, Roberta Romaniello, Elena Valente, Federica Donno, Annarita Montesanto, Paola Brovedani, Gabriele Masi, Jeffrey C. Glennon, David Coghill, Alessandro Zuddas

**Affiliations:** 1Department of Biomedical Science, Section of Neuroscience & Clinical Pharmacology, University of Cagliari, 09042 Cagliari, Italy; roberta.romaniello@gmail.com (R.R.); federica.donno87@gmail.com (F.D.); azuddas.unica@gmail.com (A.Z.); 2Child & Adolescent Neuropsychiatric Unit, “A. Cao” Paediatric Hospital-ARNAS “G. Brotzu” Hospital Trust, Department of Paediatrics, 09121 Cagliari, Italy; 3IRCCS Stella Maris Foundation, 56128 Pisa, Italy; annarita.milone@fsm.unipi.it (A.M.); elenavalente86@gmail.com (E.V.); ar.montesanto@gmail.com (A.M.); paola.brovedani@fsm.unipi.it (P.B.); gabriele.masi@fsm.unipi.it (G.M.); 4Conway Institute of Biomolecular and Biomedical Research, School of Medicine, University College Dublin, D04 V1W8 Dublin, Ireland; jeffrey.glennon@ucd.ie; 5Radboud University Medical Centre, Department of Cognitive Neuroscience, Radboud University, 6525 EN Nijmegen, The Netherlands; 6Division of Neuroscience, School of Medicine, University of Dundee, Dundee DD1 4HN, UK; david.coghill@unimelb.edu.au; 7Murdoch Children’s Research Institute, Melbourne 3052, Australia; 8Departments of Paediatrics and Psychiatry, Faculty of Medicine, Dentistry and Health Sciences, University of Melbourne, Melbourne 3052, Australia

**Keywords:** aggression, conduct disorder, oppositional defiant disorder, medications for aggression, callous-unemotional traits, D2 receptor modulators, ADHD medications, neuropsychological functioning, autonomic functioning, control design, acute placebo-controlled single-blind challenge clinical trial

## Abstract

Aggressive behaviors and disruptive/conduct disorders are some of the commonest reasons for referral to youth mental health services; nevertheless, the efficacy of therapeutic interventions in real-world clinical practice remains unclear. In order to define more appropriate targets for innovative pharmacological therapies for disruptive/conduct disorders, the European Commission within the Seventh Framework Programme (FP7) funded the MATRICS project (Multidisciplinary Approaches to Translational Research in Conduct Syndromes) to identify neural, genetic, and molecular factors underpinning the pathogenesis of aggression/antisocial behavior in preclinical models and clinical samples. Within the program, a multicentre case-control study, followed by a single-blind, placebo-controlled, cross-over, randomized acute single-dose medication challenge, was conducted at two Italian sites. Aggressive children and adolescents with conduct disorder (CD) or oppositional defiant disorder (ODD) were compared to the same age (10–17 y) typically developing controls (TDC) on a neuropsychological tasks battery that included both “cold” (e.g., inhibitory control, decision making) and “hot” executive functions (e.g., moral judgment, emotion processing, risk assessment). Selected autonomic measures (heart rate variability, skin conductance, salivary cortisol) were recorded before/during/after neuropsychological testing sessions. The acute response to different drugs (methylphenidate/atomoxetine, risperidone/aripiprazole, or placebo) was also examined in the ODD/CD cohort in order to identify potential neuropsychological/physiological mechanisms underlying aggression. The paper describes the protocol of the clinical MATRICS WP6-1 study, its rationale, the specific outcome measures, and their implications for a precision medicine approach.

## 1. Introduction

Aggression toward others is a behavior developed as part of our defense and protection [1,2]. As such, it can be considered a normal dimension of the repertoire conferring adaptive advantages contributing to survival [3]. In humans, aggressive behavior is frequent during early years: growing older, most children learn to socialize and tend to inhibit or suppress aggressive behaviors [4]; following normal morphological and functional development of the human cerebral cortex, infants learn to suppress aggressive behaviors increases while in parallel impulsive behaviors tend to diminish with increasing age, evolving into the “age of reason” [1].

Failure in suppressing aggressive behaviors and in learning effective age-appropriate self-regulation abilities may lead to pathological aggression, which can arise as an excessive response to the stimulus, causing harm to the other or toward self. Pathological aggressive behaviors often occur in individuals falling in the diagnostic category of disruptive, impulse-control and conduct disorders (DICCDs), including both conduct disorder (CD) and oppositional defiant disorder (ODD) (American Psychiatric Association, Diagnostic and Statistical Manual of Mental Disorders Fifth Edition—DSM-5, 2013) [5].

DICCDs are characterized by repetitive and persistent patterns of antisocial, aggressive, and/or defiant behavior that amounts to significant and persistent global impairment. Problems of aggression, oppositionality, and impulsivity, with or without attention deficit or hyperactivity, constitute frequent psychopathology in children and adolescents and are associated with a significant global burden [6]: they imply a worrisome impact on functioning and quality of life with strong long-term negative effects on the individual, on families and on society in general.

Aggressive behaviors are sometimes associated with callous-unemotional traits (CU). Individuals with CD showing CU traits are defined in the DSM-5 by the additional specifier “with limited prosocial emotions” (APA 2013) [5]. To qualify for this specifier, a child must have displayed in multiple relationships and settings at least two out of the following four characteristics, for at least 12 months: lack of remorse or guilt; callousness-lack of empathy; unconcern about performance (for example, at school); shallow or deficient effect (a lack of or insincere expression of feelings to others). These characteristics must reflect the individual’s typical (persistent) pattern of interpersonal and emotional functioning and not an occasional behavior [5].

Considering the poor pharmacological response of those with CU traits and the scarce availability of specific treatments [7], in the last decade, many efforts have been made to obtain a better neuropsychological characterization of subtypes of CD.

In CD, the literature suggests a strong correlation between aggressive symptoms and specific neuropsychological deficits: individuals with CD have been found to have important deficits in verbal skills and executive functions, including selective attention, cognitive flexibility, concept formation, and planning abilities [8,9,10,11]. Cognitive experimental data suggest that children with CD may also be characterized by impairments in punishment and reward processing and, more generally, in cognitive control [12,13,14] and in emotional processing [15,16]. Heart rate, skin conductance, and cortisol levels also appear to be altered in this population, in particular in children and adolescents with aggressive CD [17,18,19,20,21,22,23,24,25,26,27].

Although subjects with CD may exhibit different levels of deficits in both *cold* (i.e., working memory, response inhibition, attentional control, planning..) and *hot* (i.e., motivation, delay aversion, sensitivity to reward and punishment, emotional processing), executive functions and may show abnormal physiological parameters (such as heart rate, electrodermal activity, and cortisol levels), data on the prevalence of these characteristics are still conflicting, and the relationships between the different types of deficits have not been completely clarified. Results of the recent largest psychophysiological study to date in this field revealed no evidence for emotional under-responsiveness in CD and a very small effect for respiratory sinus arrhythmia response to sadness [28]. No difference has been found between CD subjects and controls in both sexes also on baseline heart rate, heart rate variability, and pre-ejection period, while only respiratory rate resulted higher in CD female participants [29].

Disruptive/conduct disorders are, in fact, a heterogeneous group of disorders both in terms of pathophysiology and clinical expression, and their neurobiological bases have yet to be completely clarified. Different mechanisms can lead to either CD with callous-unemotional (CU) traits with predominant instrumental aggression (low emotional reactivity, dysfunction in emotional/cognitive empathy, and deficit in decision making), or, on the other hand, to a CD characterized by reactive aggression (impulsive aggression, exaggerated effective response, deficit in processing of social-affective stimuli and in cognitive control). Blair has proposed a model delineating CD physiopathology that defines the aetiological (genetic and environmental), neural, cognitive factors associated with the behavioral aspects of CD [30]. Blair and colleagues describe the interplay of various aetiological factors and the resulting cognitive and behavioral phenotypes and define two main phenotypes: “CD with psychopathic traits” (mainly associated with decreased amygdala, striatal and ventromedial prefrontal cortex (vmPFC) reactivity, and including CU traits, antisocial behavior and instrumental behavior, and frustration-based reactive aggression) and “CD associated with anxiety and emotional lability” (mainly associated with increased amygdala reactivity, and including threat-based reactive aggression and anxiety). Both forms are likely to show under-regulated responses to social provocation. More recently, another large EU-funded study evidenced that amygdala activity in response to negative faces was not significantly influenced by aggression-related subtypes while it was decreased in the high CU traits group [31].

A crucial issue in these models is how specific neurotransmitters (noradrenaline, serotonin, and dopamine) may be involved in the modulation of the major brain areas implicated in the control of aggressive behavior (anterior cingulate cortex, prefrontal cortex, insula, amygdala, striatum, and hypothalamus) and in the underlying neuropsychological functioning. Animal models show that even a single dose of medication can induce a significant change of brain monoamine levels modifying specific neuropsychological functioning during selected tasks [32,33,34,35,36]. The available studies on humans confirm that acute doses of different medications (all able to modulate monoaminergic systems) can have specific effects on specific neuropsychological functions by modulating the neurotransmitters that regulate those processes, independently of their actual observable clinical impact [37,38,39,40,41,42]. This indicates that the acute administration of monoaminergic drugs may modulate specific regional activities, suggesting their potentially specific role in regulating the neuropsychological functioning in aggressive subjects.

In fact, some clinical evidence suggests that the use of different medications may reduce aggressive symptoms in CD. Evidence confirms that stimulant medications can improve aggression when ADHD co-occurs with disruptive behavior disorders [43,44] and in subjects with a primary diagnosis of CD [45]. Antipsychotics are the most used medication for managing aggressive behaviors in clinical practice, with risperidone being the most studied and effective within this category [46,47,48,49,50]. In addition, mood stabilizers [51,52,53,54] and other agents have been proved to exert a certain control on aggression. However, data are limited and often contradictory: no clear indications on the effectiveness of treatments depending on the specific subtype of aggression and their underlying neuropsychological mechanisms have yet been formulated [7].

On the basis of the current literature, further research is required to understand how aggressive CD/ODD patients differ from neuro-typical subjects, and exploring the putative correlations between aggressive behaviors, specific neuropsychological functions, and the physiological measures of the autonomic nervous system activity is required and may provide important information on the biological mechanisms underlying the different subtypes of aggression.

In order to enhance current knowledge on the mechanisms of pathological aggression/antisocial behaviors and their treatment, the European Commission, through the FP7-HEALTH-2013-INNOVATION-1 program, funded the European MATRICS project (Multidisciplinary Approaches to Translational Research In Conduct Syndromes), including different studies finalized to identify neural, genetic and molecular factors involved in the pathogenesis of aggression/antisocial behavior in preclinical (animal) models and clinical samples (stratified for the presence of CU traits), and proof-of-concept clinical studies aimed to define the effects of medication on specific neuropsychological domains.

Considering the limited evidence on the efficacy of medication on the treatment of aggression in the ODD/CD population, the WP6-1 study “The neuropsychological characterization of aggressive behaviour in children and adolescents with CD/ODD” was designed to highlight the neuropsychological differences between aggressive youths with CD/ODD and healthy subjects and to investigate the acute effects of a single dose of different medications on multiple neuropsychological domains, as well as their effects on the autonomic functions that are known to be impaired in these patients [19]. This may allow the identification of cognitive and physiological pathways underlying the disorders and improve the strategies for the management of aggression in clinical practice.

### 1.1. Aims of the MATRICS WP-6-1 Study

#### 1.1.1. Primary Objectives

The primary objective of the study is to compare the neuropsychological and autonomic functioning in children and adolescents with a diagnosis of ODD or CD who have clinically relevant levels of aggression with that of typically developing (TD) controls.

We primarily aim to explore if conduct problems and aggression symptoms are related to specific neuropsychological profiles (attention, working memory, decision making and risk taking, social cognition, delay aversion, emotional processing, motivation, cooperation, reward-punishment sensitivity), and autonomic functioning (heart rate, skin conductance, salivary cortisol).

#### 1.1.2. Secondary Objectives

As a secondary objective, we aim to investigate the acute effects of medications (known to impact positively on aggression in the context of CD/ODD) on specific neuropsychological and physiological features. For this purpose, we specifically explore the responses to an acute medication challenge by the administration of a single dose of a stimulant with action on monoamine reuptake (methylphenidate), a nonstimulant Serotonin and norepinephrine reuptake inhibitor (SNRI) (atomoxetine) and two antipsychotic medications with partially different mechanisms (risperidone, a full D2, and 5-HT2A antagonist; aripiprazole, a partial D2 agonist).

A further secondary objective is to evaluate the moderating or modulating role on both the neuropsychological/autonomic profile and the corresponding medication effect of socio-demographic and clinical variables such as comorbidities, SES, age and sex, previous medication, source of information (patient, parents, teacher, clinician evaluation), family structure, presence of CU traits.

Details of the protocol of the clinical MATRICS WP6-1 study are described below, including the specific outcome measures and their implications for future precision medicine approaches.

## 2. Materials and Methods

### 2.1. Study Design

This is a multicentre, phase II, control design, and acute, placebo-controlled, single-blind, challenge clinical study, employing the following 3 study periods (Table 1, see Section 2.4 study procedures for details):Phase I: a screening and clinical assessment visit;Phase II: a case-control design;Phase III: a randomized, single-blind, placebo-controlled, single-dose, cross-over, acute medication challenge.

This study (EudraCT registration number: 2015-001916-37) was conducted at two Italian sites (Università degli Studi di Cagliari and IRCCS Stella Maris, Pisa) in accordance with the Declaration of Helsinki and the good clinical practice (GCP) parameters. The study protocol, informed consents, and any other appropriate documents were submitted to the competent authority AIFA (Agenzia Italiana del Farmaco) and to the local ethical review boards (Comitato Etico Indipendente of the Cagliari University Hospital, Comitato Etico per la Sperimentazione Clinica of the Tuscany Region). AIFA approval was obtained on 4 July 2017. The ethical review board(s) reviewed the protocol as required. Before the first subject was enrolled in the trial, all ethical and legal requirements were met, and the study was approved on 29 December 2017 for the Cagliari site and on 22 January 2018 for the Pisa site.

**Table 1 brainsci-11-01639-t001:** Single-blind, placebo-controlled, acute dose, cross-over, randomized medication challenge.

Baseline Assessment ^1^ and Randomization ^2^ Group A/B	Randomization Drug/Placebo Sequence	Acute Challenge		
Week 0	Week 1	Week 1	Week 2	Week 3
Aggressive ODD/CD **Group A** (N = 60)	Group A1	Placebo	Drug A	Drug B
	Group A2	Drug B	Placebo	Drug A
	Group A3	Drug A	Drug B	Placebo
Aggressive ODD/CD **Group B** (N = 60)	Group B1	Placebo	Drug C	Drug D
	Group B2	Drug D	Placebo	Drug C
	Group B3	Drug C	Drug D	Placebo
Controls (N = 40)	No further follow up			

^1^ ODD/CD patients and TD controls. ^2^ Only ODD/CD patients. Group A: received a single dose of a stimulant (Drug A), a single dose of antipsychotic (Drug B), and placebo, each one in a different week, according to their allocation to group A1, A2, or A3. Group B: received a single dose of nonstimulant (Drug C), a single dose of antipsychotic (Drug D), and placebo, each one in a different week, according to their allocation to group B1, B2, or B3. Drug A = MPH; Drug B = Aripiprazole; Drug C = ATX; Drug D = Risperidone.

### 2.2. Study Population

#### 2.2.1. Population

The target population was 120 ODD/CD children (age 10–17 years and 10 months at screening visit; e.g., 50% 10–14 and 50% 15–17) and 40 TD controls. ODD/CD participants were inpatient or outpatient or referred by other centers. Our aim was to obtain a representative sample including both male and female subjects, considering that CD occurs in approximately 2.5% of boys and 1.5% of girls (Rowe et al., 2010). The TD group was matched to the ODD/CD group based on age, sex, and IQ.

The planned number of participants was supposed to be split equally between two Italian sites: Università degli Studi di Cagliari and IRCCS Stella Maris, Calambrone, Pisa.

#### 2.2.2. Inclusion Criteria

In order to be eligible to participate in this study within the **ODD/CD group**, patients should have an IQ ≥ 80 and comply with DSM-5 criteria for a diagnosis of ODD or CD, documented by the semi-structured K-SADS-PL interview; also, they must show significant levels of aggression, measured by a T score of ≥70 at the aggression or delinquency subscale of the Teacher Report Form (TRF), Youth Self Report (YSR), or Child Behavior Checklist (CBCL); or a score of ≥27 on the Nisonger-CBRF D-Total (a total composite score for disruptive behavior (D-Total) derived from the Oppositional and the Conduct Problems subscales). Subjects with a primary diagnosis of schizophrenia-related disorders, bipolar disorder, autistic spectrum disorder, depression or anxiety, and subjects having any psychotropic medications within the last six months before screening visit were excluded.

Subjects could be enrolled in the **TDC group** if they were the same age range, intelligent (IQ ≥ 80), drug-naïve for psychotropic medications, and with aggression below the clinical range. Subjects were excluded if they presented any psychiatric condition.

The rest of the specific inclusion criteria and the exclusion ones are listed in Table 2 and Table 3.

#### 2.2.3. Sample Size Calculation

Our approach to calculating sample size for this study is two-fold.

As per protocol, in order to analyse the case-control contrast on the dependent variables, setting alpha at 0.05 (two-tailed), we calculated that a sample size of 120 ODD/CD cases and 40 TD has 80.5% statistical power to detect a group difference with an effect size 0.30, allowing to include the covariates (sex, site, children vs. adolescents, IQ, comorbidity with ADHD). However, hypothesizing an effect size at 0.35 with a variance of 0.32, the 80.5% statistical power to detect group differences with a sample size of 88 ODD/CD subjects and 40 TD would be preserved.

For the acute medication challenge phase of the study, setting alpha at 0.05 (two-tailed), a sample size of 52 participants in each of the two studies has 80% power to detect a difference between the different conditions with an effect size of 0.4.

### 2.3. Recruitment

CD/ODD group participants were inpatient or outpatient or clinical referrals from community centers; TD controls were enrolled on a voluntary basis referred from schools or other clinical departments. TD controls were also recruited within the families of the ODD/CD group, with investigators asking the parents if they would agree to nominate a cousin or other family members or a classmate of their children who may be willing to join the study. The nominated person’s parent was asked to contact the study team, who provided information leaflets and proposed participation.

The planned number of participants was split between two Italian sites: Università degli Studi di Cagliari and IRCCS Stella Maris, Pisa.

### 2.4. Study Procedures

Study procedures and their timing are summarized in the study schedule of events (Appendix A).


**Phase I = Screening and clinical assessment visit.**


All subjects (CD/ODD and TD controls) visited the study site for a screening evaluation (Visit -1). During this visit, before starting any procedures, the study was explained to the patient and his or her parent or legal guardian, who then personally signed and dated the informed consent and assent documents. Patients underwent psychiatric screening assessments, and all criteria for enrollment were verified.


**Phase II = Case-control design.**


CD/ODD children/adolescents and TD controls underwent a baseline assessment (Table 4) at the study site with their parent/guardian for two subsequent days (Visit 0a and 0b) in order to complete the clinical and neuropsychological testing battery. Neuropsychological testing was split up into 3 sessions. The first session included 5 tasks and lasted about sixty minutes; it was administered during the first of the two days planned for baseline assessment. The second session, composed of 4 tasks and lasting about 50 min, was administered during the second of the two days planned for baseline assessment. The third session, composed of 4 tasks and lasting about 50 min, was administered 45 min after the end of the second session to allow for resting time.

During these baseline visits, participants were also assessed on selected autonomic measures (heart rate, skin conductance, salivary cortisol) at rest and during the testing session.


**Phase III = Randomized, single-blind, placebo-controlled, single-dose, cross-over, acute medication challenge.**


CD/ODD children/adolescents were enrolled in one of the two randomized, single-blind, placebo-controlled, single-dose, cross-over, acute challenge arms (Table 1). The control cohort did not take part in this phase. Each subject was randomly exposed to a single dose of the drug each week for three consecutive weeks according to the condition of their randomization group (A or B). Patients randomized to group A received a single dose of a stimulant, a single dose of antipsychotic medication, and a single dose of placebo, each one in a different week (Visit 1, 2, and 3), according to the order of their allocation to group A1, A2 or A3. Following the same procedure, patients randomized to group B received a single dose of a nonstimulant, a single dose of antipsychotic medication, and a single dose of placebo, each one in a different week, according to their allocation to group B1, B2, or B3.

After the administration of a single dose of one of the four selected medication (methylphenidate, atomoxetine, risperidone, aripiprazole) or placebo, patients underwent a subset of the tasks performed during the baseline assessment (only the second and the third session tasks, for a total task time of about one hour and forty minutes; Table 4).

During this phase, participants were also reassessed on selected autonomic measures (heart rate, skin conductance, salivary cortisol) at rest and during the testing session as in the baseline assessment.

We controlled for practice effects by using a modified Latin square design, using parallel versions of tasks where available, and having a one-week medication-free break between testing sessions.

#### 2.4.1. Method of Assignment to Treatment

At the baseline visit, CD/ODD patients were randomly assigned to Group A or Group B: each subject was given a precise sequence of drug administration within the group (e.g., A1/A2/A3) as shown in Table 1. Assignment to medication groups was determined by a computerized randomization list generator, set by the pharmacist from the Cagliari University hospital, not directly involved in patient screening and assessment.

To ensure groups were balanced among sites, the randomization had been stratified by age and sex.

#### 2.4.2. Blinding

During the acute single-dose medication challenge study, at each testing session, the patient was unaware of the specific nature of medication or placebo tablet (single-blind phase; Study Period III). Investigators were unblinded to the patient’s treatment.

The single-blind design and the unblinding of the investigator will not affect the outcome of the study as the task administration is automated with set scripts, and outcome measures are objectively determined by the computer.

#### 2.4.3. Concomitant Therapy

No concomitant psychotropic medication (i.e., psychostimulants, antipsychotics, SNRI, mood stabilizers, and antidepressant medication) was allowed during baseline visit or study period III, nor permitted during the previous six months. If the patient was on any other medication for any chronic condition, the investigator, according to his clinical judgment and upon consent by the patient and patient’s parent, could allow the co-administration of the two drugs or suggest tapering off, stopping, or completely washing out the concomitant medication six months before the baseline visit.

#### 2.4.4. Screening and Clinical Assessment

The screening and clinical assessment session (I) included the following evaluations:-*K-SADS-PL* [55]: a semi-structured diagnostic interview assessing psychopathology based on DSM-IV categories. This interview was used to assess the primary diagnosis and the possible comorbid conditions, including substance (ab)use. The presence/absence of an abuse of tobacco, alcohol, and/or illegal psychotropic drugs was investigated, and the frequency of use during the last 12 months was recorded separately within the CRF.-*Wechsler Intelligence Scales*: To obtain an estimate of IQ, WISC-IV (Wechsler Intelligence Scale for Children—Fourth Edition) [56] or WAIS-IV (Wechsler Adult Intelligence Scale) [57] was administered, depending on age, for an estimate of global intelligence functioning.-*Questionnaires:*Child Behavior Checklist (CBCL), Teacher Report Form (TRF) Youth Self Report (YSR) [58]: these questionnaires are part of the Achenbach System of Empirically Based Assessment (ASEBA) and provide a measure of general functioning as well as internalizing and externalizing problems.Conners’ Parent Rating Scale-Revised Short Form (CPRS-RS) [59]: an abbreviated version of the factor-derived subscales that assess a cross-section of ADHD-related symptoms and problem behaviors. The parent or caregiver typically responds on the basis of the subject’s behavior over the past month.Behavior Rating Inventory of Executive Function (BRIEF) [60]: an 86-item questionnaire fulfilled by parents on executive function behaviors at home and at school for children and adolescents ages 5–18.Inventory of Callous-Unemotional Traits (ICU) [61]: a 24-item questionnaire administered to parents designed to provide a comprehensive assessment of CU traits and including three subscales (Callousness, Uncaring, and Unemotional).-*Rating scales:*Modified Overt Aggression Scale (MOAS) [62]: a short, widely used rating instrument for assessment of verbal aggression, aggression against property, auto-aggression, and physical aggression. It needs to be administered by the clinician to the parent/caregiverThe Nisonger Child Behavior Rating Form (NCBRF-TIQ) parent version: a 66-item measure used to assess child and adolescent behavior in children with disruptive behavior disorder; it needs to be administered by the clinician to the parents/caregiver [63].-*Other:*The Clinical Global Impressions-Severity (CGI-S) [64]: a single-item rating of the clinician’s assessment of the severity of symptoms in relation to the clinician’s total experience (Guy, 1976; NIMH, 1985). Severity is rated on a 7-point scale (1 = normal, not at all ill; 7 = among the most extremely ill subjects).The Children’s Global Assessment (C-GAS) [65]: a global, one-dimensional clinician rating of social, family, academic and psychiatric functioning. Scores on the measure range from 1 (most impaired, persistent risk to hurt) to 100 (healthiest; no symptoms).

#### 2.4.5. Risk Assessment for Pharmacological Treatments (Patients Only)

The screening session (I) included the following evaluations to assess the risk of cardiological contraindications and the risk of cardiac adverse events:-Assessment of the history of exercise syncope, undue breathlessness, and other cardiovascular symptoms;-Heart rate and blood pressure;-Family history of cardiac disease and examination of the cardiovascular system;-An ECG, performed within the last six months;-A cardiological consultation if considered necessary by the investigator.

A check of previous blood chemistry (within the last six months), exhaustive medical history, and a physical and neurological examination were performed in order to exclude any contraindication to the use of psychotropic medications.

#### 2.4.6. The Neuropsychological Task Battery

Three of the tasks used in this study, which mainly assess “cold” executive functions (set-shifting ability, sustained attention, and working memory), are from the Cambridge Neuropsychological Test Automated Battery (CANTAB RESEARCH SUITE https://www.cambridgecognition.com/cantab/ (accessed on 1 December 2021) [66]), a well-known and validated cognitive software. CANTAB includes highly sensitive, precise, and objective measures of cognitive function correlated to neural networks and is widely used in the clinical research field.

The other ten tasks, mainly assessing “hot” executive functions, are instead derived from EMOTICOM [67], an innovative and not yet commercialized neuropsychological battery developed by the same research group that developed the CANTAB. These tasks assess 4 core domains of affective cognition (emotion processing, motivation, impulsiveness, and social cognition).

All computerized tests were administered using a touchscreen tablet (10.1-inch screen). Apart from the first one (motor screening (MOT) from Cantab), the other tasks were administered on a random sequence for each subject in order to minimize any impact of fatigue across the study. During Phase III, each subject belonging to the CD/ODD group was re-tested by using the same sequence he/her was administered during the baseline assessment.


**Baseline assessment (First session) neuropsychological tasks:**
-Motor screening (MOT)A simple reaction time screening test from the CANTAB battery, the first one to be administered. It aims to familiarize the subject with the test material and to assess the presence of any limitations in the use of the device (vision problems, hearing problems, motors, etc.).-Intra-Extra Dimensional Set Shift (IED):IED (Cantab) is a test that assesses visual discrimination and attentional set formation, maintenance, shifting, and flexibility of attention.*Outcome measures*: This test has many outcome measures; the main ones include accuracy (total errors) and shift ability (numbers of stages completed).-Face and Eyes Emotional Recognition Task (FEERT)The FEERT (Emoticom) measures the ability to identify emotions in facial/eyes expressions. Target emotions are presented using ten different images for each of the four basic emotions (happiness, sadness, anger, and fear), each showing different levels of intensity. Choice accuracy and latency are recorded.*Outcome Measures*: The outcome measures are accuracy across emotions and intensities, overall response latencies (mean reaction times), and effective bias.-Delay Discounting (DD)DD (Emoticom) is a measure of inhibition/impulsivity and delay aversion that assesses the rate of discounting across delays and probabilities.*Outcome Measures*: Area under the curve (AUC) and k calculated from indifference points.-Moral Judgment (MJ)MJ (Emoticom) uses cartoon figures to depict moral scenarios; it assesses normative emotional reactions to being a victimizer (agent) or a victim in the moral situation.*Outcome Measures*: The outcome measures of the task are ratings for the four emotions (guilt, shame, annoyance, and good/bad), which can be looked at across all conditions: agent/victim condition (situations in which the subject is asked to identify himself with the victimizer or with the victim, respectively) combined with intentional/unintentional condition (situations in which an intentional or accidental harm is acted) in order to explore the effect of intention upon moral emotions in moral scenarios.-Prisoner Dilemma (PD)PD (Emoticom) assesses cooperation.*Outcome Measures*: Split and steal behavior across three different opponents (each with a different strategy: aggressive (tit for tat but starts with steal), tit for two tats (starts with split, then changes behavior after the player stolen two times consecutively) and a cooperative player who always splits) and for each type of contribution (win, lose, draw). Response latency is also an outcome variable.



**Baseline (second session) and each medication trial visit assessment neuropsychological tasks:**
-Rapid Visual Information Processing (RVP)RVP (Cantab) is a measure of visual sustained attention.*Outcome measures*: The nine RVP outcome measures cover latency (mean reaction times), response accuracy, and target sensitivity.-Delayed Matching to Sample (DMtS)DMtS (Cantab) assesses forced-choice recognition memory for non-verbalizable patterns, testing both simultaneous matching and short-term visual memory.*Outcome measures:* This test has several outcome measures, the main ones assessing latency (the participant’s speed of response) and the proportion of correct patterns selected when the patterns are presented simultaneously or after brief delays from the stimulus.-Progressive Ratio Task (PRT)PR Task (Emoticom) assesses participants’ motivational ‘breakpoint’.*Outcome measures*: main measures are trials completed, post-reinforcement pause (average time taken to initiate the next trial following a reward), and running rate (time taken to complete the block of trials).-Face Affective Go/NoGo Task (FAGNG)FAGNG (Emoticom) assesses information processing biases for positive and negative facial expressions (biased emotional attention).*Outcome Measures*: The task records proportions of hits, misses, correct rejections, and false alarms. This is used to calculate accuracy, reaction times, and effective bias across conditions.-New Cambridge Gambling Task (New CGT)NCGT (Emoticom) assesses decision-making and risk-taking behavior and investigates reward-seeking and punishment avoidance separately.*Outcome Measures*: The six outcome measures cover risk taking, quality of decision making, deliberation time, and overall proportion bet, split in loss condition, and win condition.-Reinforcement Learning Task (RLT)RLT Task (Emoticom) assesses reward and punishment sensitivity.*Outcome Measures*: Outcome measures include learning rate and response times.-Theory of Mind (TOM)TOM (Emoticom) assesses information sampling in socially ambiguous situations.*Outcome Measures*: Information sampling, preference for feelings, thoughts, and facts (proportion of faces/thoughts/facts selected by the subjects to help resolve ambiguity), outcome choice (negative, positive, or neutral), and outcome choice confidence.-Ultimatum Game (UG)UG (Emoticom battery) assesses fairness sensitivity and punishment tendency.*Outcome Measures:* Outcome measures include “accept” percentage for each level of opponent offers (90%, 80%, 75%, 70%, 65%, 60%, or 50%) used to assess risk adjustment.


#### 2.4.7. Physiological Measures

-*Saliva cortisol samples collection*: Participants were asked to collect saliva using a “passive drool” method in order to measure salivary cortisol. The saliva cortisol samples were collected during V0a and V0b visits and, for the CD/ODD sample only, during each subsequent visit (V1, V2, V3). Each visit was scheduled in the morning. One baseline sample and one stress sample were collected before and after the testing session, respectively. The baseline sample was collected before the testing session in a time window between 8:00 and 9:30 a.m. while the stress sample was collected at the end of the testing session (exact time was supposed to vary based on the time needed for the execution of the whole neuropsychological battery for each subject). The sampling time was recorded in the patient’s CRF. If the participant had trouble spitting, sugar- and flavor-free chewing gum were provided to assist salivation. They were asked to rinse their mouths with water and then waited approximately 1 min before producing each sample. All samples were centrifuged after collection and then frozen and stored at −20 °C until assay. Cortisol levels were assessed by an external lab (Ospedale San Raffaele, Milano);-*Autonomic measures by using Empatica E4*: During the performance of all above-mentioned tasks, subjects underwent an autonomic profile measurement by a wristband able to record heart rate (HR) and heart rate variability by a photoplethysmography technique. HR was recorded for five minutes while the participant was at rest to yield baseline and continuously during the performance of the whole neuropsychological battery. The same wristband allowed to record the electrodermal activity in terms of skin conductance (referred to as galvanic skin response), arousal, and excitement.

#### 2.4.8. Vital Signs, Body Temperature, Height, and Weight

Fifteen minutes before the beginning and 15 min after the conclusion of the neuropsychological testing sessions, vital signs (blood pressure, heart rate) were recorded. Body temperature, height, and weight were recorded, and physical examination was also performed.

#### 2.4.9. Single Acute Medication Administration

Within the single-blind, randomized, placebo-controlled, single-dose, cross-over medication challenge, the ODD/CD subjects were randomly assigned to have a single dose of placebo or two of the following medications once a week for three consecutive weeks: methylphenidate, atomoxetine, risperidone, aripiprazole.

The investigator or his/her designee was responsible for explaining the characteristics of the investigational agent(s) and possible adverse effects to the patients and his/her parents/legal guardian, maintaining accurate records of study drug dispensing at each visit.

Patients received a single dose of medication at the investigator study site one hour and a half before the beginning of the first testing session (e.g., 8.00 a.m. if the testing session is supposed to start at 9:30 a.m.). The time of administration had been estimated as the best time in order to evaluate the change of the performance on the various tasks during the pharmacodynamics window for these medications.

Dosages of each medication had been chosen according to the available literature considering the safety and tolerability of each drug on a single administration. Single doses were assigned to the participants according to their weight range (Table 5).

### 2.5. Outcome Measures: Study Parameters/Endpoints

#### 2.5.1. Main Outcome Measures

Main outcome measures include the following:−Quantitative and qualitative measures from the neuropsychological tasks: reaction times (response latency), accuracy (number/percentage of errors), test completion, learning rate, motivation, cooperation, reward-punishment sensitivity on the neuropsychological tasks from CANTAB and EMOTICOM batteries;−Physiological measures: heart rate, skin conductance, and salivary cortisol levels at rest and during test performance.

#### 2.5.2. Secondary Study Parameters/Endpoints

Secondary endpoints include measures to investigate the association between severity, type of aggression, and performance on the neuropsychological tasks:−Screening questionnaires: TRF, YSR, CBCL, CPRS, BRIEF, ICU;−Screening rating scales: MOAS and Nisonger interview;−CGI-S, C-GAS.

### 2.6. Handling and Storage of Data and Documents

Data were handled confidentially and anonymously. A subject identification code list was used to link the data to the subject. The code was not based on the patient’s initials and birth date. The key to the code was safeguarded by the investigators. Demographic data (name, address, etc.) and identification numbers were coupled in a file, which was saved on a pc protected by a password, only accessible to the investigators. The handling of personal data complies with Personal Data Protection Acts.

Salivary samples were received by the laboratory, which performed the analyses. Samples were given a unique code, consisting of the type of sample, the visit and the date of collection, and a unique, consecutive number.

### 2.7. Statistical Analysis

#### 2.7.1. Primary Study Parameter(s)


*Cognitive-behavioral measures*


For each task, mean reaction time, accuracy rate, and other quantitative parameters for every participant will be calculated.

Chi-square or one-way analysis of variance (ANOVA) tests will be used to assess group differences in case of data meeting assumptions of normality and homogeneity of variance, while covariates will be explored by using analysis of covariance (ANCOVA) and, thereafter, by determination of simple effects or interactions. All other data will be compared using appropriate non-parametric tests (e.g., Mann–Whitney U test) or by using a bootstrap-based non-parametric ANOVA. Simple and multiple regression models will be applied on the whole sample and on the CD/ODD group with the descriptive measures of CU traits and aggression as dependent variables and all outcome measures as the main predictors. Subjects with an ICU score ≥32 will be considered as having high CU traits based on previous studies where a cut-off score of 32 on the ICU was used to split groups of subjects with psychopathic tendencies [68], CD [69], or non-CD diagnosis [70].

Covariates will include age, sex, IQ. The primary diagnosis of ODD or CD separately for a between-group comparison as well as a binary covariate will be explored within the total sample; comorbidities as ADHD, anxiety, and depression, etc., will also be analyzed as covariates.

To explore the potential effects of medication, data from the medication acute challenge will be entered into a bootstrap repeated-measures ANOVA for analysis.

Neuropsychopharmacological response to medication is defined as the score change on the primary measures from each task.

Correlations, simple and multiple regressions, and ANCOVAs will be used to investigate demographic, clinical, neuropsychological, and neurophysiological predictors of acute pharmacological response to establish their role as moderating or modulating variables and to define the correlation between severity, CU traits, and neuropsychological/autonomic profiles.


*Autonomic measures*


If, as expected, the raw cortisol values are positively skewed, they will be normalized using a log transformation. To assess group differences in cortisol, HR, and skin conductance during task performance, repeated-measures ANOVAs will be performed with the group as a between-subjects factor and session as a within-subjects factor. To quantify cortisol, HR, and skin conductance responsiveness to the stress related to the performance during tasks, change relative to baseline will be calculated. With regard to cortisol levels, the collection time of the samples will be used as a covariate. A one-way ANOVA will be used for group comparisons.

#### 2.7.2. Secondary Study Parameters

Chi-square or one-way ANOVA tests will be used to assess group differences in demographic variables, as appropriate.

We will use (I) ANOVA to compare the ODD/CD group and TD group on all outcome measures and (II) multiple regression models on the whole sample (N = 160) with the descriptive measures of CU traits, aggression, and conduct problems as dependent variables, and all outcome measures as the main predictors. Covariates will include age, sex, SES, IQ, previous medication, ADHD symptoms, source of information, family structure.

## 3. Discussion and Clinical Implications

ODD and CD have significant long-term implications for affected patients and their families and represent a significant public health problem. While there is some evidence to support the use of medications in reducing conduct problems, such evidence across adolescent development is limited and contradictory [71]. Clear indications on the effectiveness of treatments for specific subtypes of aggression have not yet been formulated, with many clinicians prescribing medications off-label. Currently, in randomized controlled trials, a relatively large pharmacological effect on aggression has been reported for risperidone [72] and methylphenidate [43]. There is some limited evidence for clinical efficacy in CD for other D-2 modulators, including aripiprazole, and there is some low-quality evidence, mainly derived from open-label trials, to support a small effect of mood stabilizers. Only a few studies investigated patients with CD as the primary diagnosis, and very few discriminated between different types of aggression or reported measures of CU traits, thus providing inconclusive results on the modulating role of CU traits on the efficacy of medications [7].

Considering the poor knowledge in the field, the high heterogeneity of the two disorders, and the fact that the neurobiological bases of disruptive/conduct disorders have not been completely clarified, the present study was designed to develop a better understanding of the neuropsychological and autonomic characteristics underpinning conduct and oppositional defiant disorder and to identify specific pharmacological responses for the development of novel targeted pharmacological treatments.

Previous single-dose cross-over studies in children and adolescents on stimulant medications have been performed in patients with ADHD, showing important effects of MPH on various aspects of cognition beyond clinical symptoms [39]. These data suggested that the administration of single doses of medication can have immediate effects on neurobiological correlates and that clinical and neuropsychological effects may not necessarily coincide, with consequent important implications for treatment planning.

In the present study, we include subjects with a primary diagnosis of CD or ODD and aggression, thus trying to trace specific phenotypes for ADHD and for disruptive/conduct disorders, despite the frequent overlap between the two disorders in the clinical population. The high comorbidity with ADHD and the impact of inattention and impulsiveness on the global impairment of CD/ODD subjects can, in fact, explain the severity of the disease and the predisposition to the persistence of antisocial behaviors [73,74]. Based on this understanding, several clinical variables (including ADHD comorbidity, family history of psychiatric disorders, or previous treatments) will be explored to investigate their effects on predictive models for conduct problems and aggression. In particular, the role of inattention in mediating some of the executive function deficits known to be present in CD/ODD subjects [75,76] will be investigated.

Apart from ADHD core symptoms, another complicating factor in the management and treatment of CD is related to the presence of CU traits. Although many youths with CD and CU traits seem to respond to treatment, most studies have found that CU traits predict relatively poor treatment outcomes, independently of conduct problem severity before treatment [77,78]. An advantage of the present protocol is the possibility of investigating if and how CU traits may modulate the effects of medications, considering that a poorer response for patients with high CU traits may be expected [77].

Considering the absence of drugs registered for this specific indication (aggression) in this particular population (pediatric ODD and CD with normal IQ), the enrollment into the present study represents an important opportunity to strictly monitor these patients and stimulate their awareness and compliance to therapeutic interventions. Moreover, besides the effects of the single doses of the medications on the outcome measures (positive and negative effects, such as cognitive impairment), the data collected on adverse events may also be of particular interest, considering the impact that the appearance of side effects usually has on patient compliance with chronic treatments.

### 3.1. Strengths and Limitations of the Present Study

Considering the current lack of evidence on pharmacological effects on cognitive functioning in the ODD and CD population with and without CU traits, the main strength of the present protocol is to investigate the acute effects of different medications on different neuropsychological functions within the same study, as well as the medication effects on the autonomic functions that are known to be impaired in these patients. This may allow us to verify the putative role of cognitive and physiological pathways underlying the disorder in order to improve the knowledge on aggression management. The information derived from the modulation of different monoaminergic systems (noradrenergic, serotoninergic, and dopaminergic) by different medications could be crucial for personalized medicine strategies and the development of more effective treatments.

The main limitations of this study protocol include that the Emoticom tasks are not yet standardized in a pediatric population and that the battery administered in this study is an “experimental” version. Furthermore, some of the Emoticom tasks include reading words (especially Theory of Mind and Moral Judgment), which could be less suitable for younger patients or those with specific reading disabilities, therefore requiring additional effort, which might interfere with their performances. Another important limitation is that while the study protocol includes a measure of CU traits (the ICU questionnaire), no specific measures for impulsive vs. proactive aggression assessment have been included: we are aware that their inclusion would have clarified the association between subtypes of aggression and CU traits, possibly adding useful information on drug effects profiles.

### 3.2. Benefits and Risks for Participants

The present study aims at collecting scientific information rather than assessing immediate therapeutic benefits for patients: no efficacy on clinical symptoms was foreseen from the administration of single doses of the study drugs, and patients were not provided chronic pharmacotherapy within the study.

Medications used in the present protocol are indicated for comorbid disorders: and, with the exception of risperidone (registered for behavioral problems in children and adolescents with autism or intellectual disability), are not currently specifically indicated for the treatment of aggression for either conduct disorder or oppositional defiant disorder: the benefits for the enrolled patients could be related to the possibility of identifying their potentially most effective treatment on the basis of their pharmacological response to the neuropsychological tasks and of their autonomic profile.

## 4. Conclusions

Validating the hypothesis that specific neuropsychological and autonomic features found in CD/ODD aggressive children and adolescents are responsive to an acute dose of medication may lead to a specific targeted precision medicine approach for CD with and without CU traits.

Based on the results of this study, the more promising medications may be studied in future prolonged, controlled trials to test their efficacy in modifying cognitive and behavioral symptoms as well as the clinical progress of the psychiatric disorder. Taken together, the results of the present study may contribute to the development of specific guidelines for selecting appropriate medications according to patients’ aggressive profiles.

Finally, the results of this study could lead to the definition of a neuropsychopharmacological toolkit, which would include the tests found as more useful and sensitive for predicting efficacious responses to medications to be used in clinical practice.

## Figures and Tables

**Table 2 brainsci-11-01639-t002:** Inclusion and exclusion criteria for ODD/CD group.

Inclusion Criteria
IQ ≥ 80 (Wechsler intelligence scale, within the last two years before enrollment);
Aged between 10 and 17 years and 10 months at the screening visit;
Diagnosis of ODD or CD, based on the DSM-5 criteria, documented by the semi-structured K-SADS-PL interview; patients meeting criteria for comorbid ADHD, depression, anxiety, or PTSD (as to the clinical judgment of the investigator) will not be excluded from study participation;
Significant levels of aggression, measured by a T score of ≥70 at the aggression or delinquency subscale of the Teacher Report Form (TRF), Youth Self Report (YSR), or Child Behavior Checklist (CBCL); or a score of ≥27 on the Nisonger-CBRF D-Total (a total composite score for disruptive behavior (D-Total) derived from the Oppositional and the Conduct Problems subscales);
Eligibility to be treated with a pharmacological therapy based on previous medical and instrumental cardiological assessments and based on previous blood chemistry (performed within the last six months), current physical and neurological examination;
Drug-naïve for psychotropic medications (psychostimulants, antipsychotics, SNRI, antidepressants, mood stabilizers) or off any psychotropic medication within the last six months;
Sexually active and of childbearing-potential subjects (WOCBP: women of childbearing potential) must have a negative urine pregnancy test at the screening visit and at the baseline visit, and at each week, during the acute challenge phase of the study;
Subjects’ parents/legal guardians must provide and sign informed consent documents; patients must provide informed consent and sign consent or assent documents.
**Exclusion Criteria**
Primary DSM-5 diagnosis of schizophrenia-related disorders, schizophrenia, bipolar disorder, autistic spectrum disorder, depression or anxiety;
Any psychotropic medications (psychostimulants, antipsychotics, SNRI, antidepressants, mood stabilizers) within the last six months before screening visit;
The subject is pregnant or nursing;
Body weight < 30 Kg;
Any acute or unstable medical condition that, in the opinion of the investigator, would compromise participation in the study;
History of severe allergies to medications, in particular hypersensitivity to neuroleptics, or of multiple adverse drug reactions, or the patient has any contraindications to the use of methylphenidate, atomoxetine, risperidone, or aripiprazole.

**Table 3 brainsci-11-01639-t003:** Inclusion and exclusion criteria for the TDC group.

Inclusion Criteria
IQ ≥ 80 (Wechsler intelligence scale, within the last two years before enrollment);
Age between 10 and 17 years and 10 months at the screening visit;
Aggression below the clinical range, T < 70 on the aggression or delinquency subscale of the Teacher Report Form (TRF), Youth Self Report (YSR), Child Behavior Checklist (CBCL), and score of <27 on the Nisonger-CBRF D-Total (composite of Disruptive Behavior Disorder subscales);
Drug-naïve for psychotropic medications;
Subjects’ parents/legal guardians must provide and sign informed consent documents; TD control must provide informed consent and sign consent or assent documents;
If the patient is a girl who is sexually active and WOCBP, she must have a negative urine pregnancy test at the screening visit and at the baseline visit.
**Exclusion Criteria**
A primary DSM-5 diagnosis of ADHD, ODD, CD, or any other psychiatric condition;
Any psychotropic medications (psychostimulants, antipsychotics, SNRI, antidepressants, mood stabilizers) within the last six months before screening visit;
The subject is pregnant or nursing.

**Table 4 brainsci-11-01639-t004:** Neuropsychological assessment.

Task	Battery
**First Day (Visit 0a): Testing Session (aprx 60 min)**	
Intra-Extra Dimensional Set Shift (IED)	CANTAB
Face and Eyes Emotional Recognition Task (FEERT)	EMOTICOM
Delay Discounting (DD)	EMOTICOM
Moral Judgment (MJ)	EMOTICOM
Prisoners Dilemma (PD)	EMOTICOM
**Second Day (Visit 0b, 1, 2, 3): First Session (aprx 50 min.)**	
Rapid Visual Information Processing (RVP)	CANTAB
Delayed Matching to Sample (DMS)	CANTAB
Progressive Ratio Task (PRT)	EMOTICOM
New Cambridge Gambling Task (NCGT)	EMOTICOM
**Second Day (Visit 0b, 1, 2, 3): Second Session (aprx 50 min)**	
Face Affective Go/NoGo (FAGNG)	EMOTICOM
Reinforcement Learning Task (RLT)	EMOTICOM
Theory of Mind (ToM)	EMOTICOM
Ultimatum Game (UG)	EMOTICOM

**Table 5 brainsci-11-01639-t005:** IMP dosages for specific weight ranges.

Weight (kg)	MPH (Medikinet)	ATX (Strattera)	RISPERIDONE (Risperdal)	ARIPIPRAZOLE (Abilify)
≥50	20 mg	40 mg	1 mg	5 mg
≥35–<50	15 mg	25 mg	0.5 mg	2.5 mg

## Data Availability

Data are not publicly available at the moment. They could be available on request from the corresponding authors.

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
