# Peer review of "Neuropsychological Characterization of Aggressive Behavior in Children and Adolescents with CD/ODD and Effects of Single Doses of Medications: The Protocol of the Matrics_WP6-1 Study"

_brainsci, 2021, doi:10.3390/brainsci11121639_

Round 1

Reviewer 1 Report

Neuropsychological characterization of aggressive behaviour in children and adolescents with CD/ODD and effects of single doses of medications: the protocol of the Matrics_WP6-1 study

Carla Balia et al., and the MATRICS consortium.

Summary: This paper is based on the FP7 funded MATRICS (Multidisciplinary Approaches to Translational Research In Conduct Syndromes) project, describing the protocol of the clinical MATRICS Work Package (WP6) as paper 1 and the associated outcome measures. Specifically, highlighting the neuropsychological differences between aggressive youths with CDs and healthy subjects and to investigate the effects of different medications on neuropsychological domains, as well as their effects on the autonomic functions. This paper presents a solid and detailed account of the planned protocols and outcomes measures for that study and reads as an SOP for WP6 study 1.

Comments:

Pg. 2 (Introduction): “Pathological aggressive behaviours often occur in individuals with a diagnosis of Disruptive, Impulse—Control and Conduct Disorders, including both Conduct Disorder (CD) and Oppositional Defiant Disorder (ODD) (American Psychiatric Association, Diagnostic and Statistical Manual of Mental Disorders Fifth Edition - DSM-5, 2013)”

Can the authors please clarify if they are referring to disruptive behavioural disorders (DBD’s) here, such as the wider terminology ‘conduct problems’ or focusing on conduct disorder (CD) and oppositional defiant disorder (ODD) as defined in the DSM-5. At present, the wording is a little confusing. This should be used consistently throughout the Introduction and manuscript (e.g. reference to disruptive/conduct disorders or just CD in different places).

The authors might want to consider how grouping young people with CD differ based on high- versus low-levels of CU traits, as had been done categorically in numerous other papers, as well as explore differences in CD and ODD groups separately. The influence of co-morbid symptoms, particularly those associated with medication such as ADHD, anxiety, depression, should also be explored or at least accounted for as a covariate.

Minor

Spelling mistake: 1.1. Aims of the MATRCS WP-6-1 study

This work would benefit from reference to other work now published from other large, FP7 projects on CD, notably the FemNAT-CD (www.femnat-cd.eu) project:

Oldenhof H, Jansen L, et al.,Popma A. (2020) Psychophysiological responses to sadness in girls and boys with conduct disorder. J Abnorm Psychol. 2020 Nov 12. doi: 10.1037/abn0000524. Epub ahead of print. PMID: 33180540.

Oldenhof, H., Prätzlich, M., Ackermann... Popma, A. (2019). Baseline autonomic nervous system activity in female children and adolescents with conduct disorder: Psychophysiological findings from the FemNAT-CD study. Journal of Criminal Justice, 65, [101564]. https://doi.org/10.1016/j.jcrimjus.2018.05.011

Reviewer 2 Report

      • A brief summary

      This manuscript aims to investigate the effects of four specific (single dose) drugs in a cohort with children and adolescents with Conduct Disorder and Oppositional Defiant Disorder. It describes the protocol of the clinical multicentre (control design, and acute, placebo controlled, single blind) study, named the MATRICS WP6-1 study.

      Two aims were specified. (I) The primary objective of this study was to compare the neuropsychological and autonomic functioning in children and adolescents with a diagnosis of ODD or CD who have clinically relevant levels of aggression with that of Typically Developing (TD) controls. (II) The second objective was to investigate the acute effects of medications on specific neuro-psychological and physiological features.

      • General concept comments

      Article:

      I am worried that the inclusion of both ODD and CD in one group could make interpretation of the results difficult, since both are considered different diagnoses and might have different neurobiological bases. p.3 “Disruptive/conduct disorders are in fact a heterogeneous group of disorders both in terms of pathophysiology and clinical expression, and their neurobiological bases have yet to be completely clarified.” How will the authors investigate (and is there enough power to detect) if conduct problems and aggression symptoms are related to specific neuropsychological profiles, when the group is so heterogeneous?

      Furthermore, will this protocol take into account other factors, such as smoking, alcohol and substance (ab)use? Which are more often observed in these ODD/CD samples compared to non-ODD/CD peers, and could affect some of the proposed measurements (e.g., smoking: https://doi.org/10.1016/j.jcrimjus.2018.01.004)

      I like that a saliva cortisol samples collection is included within this protocol. However, it is unclear to me at what times these will be administered/measured. Will all participants be tested within the same time frame/window during the day? This is an important factor since cortisol varies during the day. If the measurement time differs between groups/individuals, how will the authors aim to analyze this dataset?

      p.14 “Correlations, simple and multiple regressions, and ANCOVAs will be used to investigate demographic, clinical, neuropsychological and neuro-psycho-pharmacological pre-dictors of clinical response, to establish their role as moderating or modulating variables, and to define the correlation between severity, CT traits and neuropsychological/auto-nomic profiles.”  How do the authors define the clinical response they mention on page 14?

      I am wondering whether the data is already collected, as I was in the assumption that the Matrics consortium has finished. If the data is already collected, what is the rationale to write a study protocol and not publish the results?

    • If the data is not yet fully collected, then the manuscript would benefit from stating this more clearly and rewriting the past tense to a future tense.

      The manuscript would benefit from a more detailed description and references of papers that have examined the effect of medication in samples with CD in the introduction. The authors do refer to their own meta-analyses several times, but do not include the papers used in that meta-analyses within this manuscript.

      For example:

      In this sentence references (31-34) are used for ADHD papers, and not CD/ODD papers

      p.3 “The available studies on humans confirm that acute doses of different medications (all able to modulate monoaminergic systems) can have specific effects on specific neuropsychological functions by modulating the neu-rotransmitters that regulate those processes, independently of their actual observable clin-ical impact [31-34].”

      p.3 “In fact, some clinical evidence suggests that the use of different medications (psy-chostimulants, antipsychotics, mood stabilizers and other agents) may reduce CD symp-toms, but data are limited and often contradictory: no clear indications on the effective-ness of treatments depending on the specific subtype of aggression and their underlying neuropsychological mechanisms have yet been formulated [7].”

      Here only the meta-analyses (7) are mentioned, the manuscript would benefit from adding more details within this section about the studies that investigated medication effects in CD/ODD samples.

      p.7 ‘However, hypothesizing an effect size at 0.35 with a variance of 0.32, the 80.5 % statistical power to detect group differences with a sample size of 88 ODD/CD subjects and 40 TD would be preserved.”

    • Where does this 0.35 effect size come from, is there a reference?

      Minor:

    • Some sentences/claims are missing references.

      For example: p. 15 “Currently, in randomized controlled trials, a relatively large pharmacological effect on aggression has been reported for risperidone and methylphenidate.”

      Formatting:

      There are open spaces in multiple sentences throughout the manuscript, probably a formatting error. E.g.:

      p.3 “have specific effects on specific neuropsychological               functions by modulating the”

      Table 1. “Group B: received a single dose of not stimulant (Drug C)”

      Should be ‘non-stimulant’.

Round 2

Reviewer 2 Report

The authors have revised their manuscript thoroughly.
I can say the manuscript has been sufficiently improved for publication.